# Putative *Daucus carota Capsanthin-Capsorubin Synthase (DcCCS)* Possesses Lycopene β-Cyclase Activity, Boosts Carotenoid Levels, and Increases Salt Tolerance in Heterologous Plants

**DOI:** 10.3390/plants12152788

**Published:** 2023-07-27

**Authors:** Carolina Rosas-Saavedra, Luis Felipe Quiroz, Samuel Parra, Christian Gonzalez-Calquin, Daniela Arias, Nallat Ocarez, Franco Lopez, Claudia Stange

**Affiliations:** 1Departamento de Biología, Facultad de Ciencias, Universidad de Chile, Las Palmeras 3425, Ñuñoa, Santiago 7750000, Chile; rosas.biotec@gmail.com (C.R.-S.); luis.quiroz@universityofgalway.ie (L.F.Q.); samuel.parra.a@gmail.com (S.P.); christian.gonzalez@usach.cl (C.G.-C.); danielaloreto.arias@gmail.com (D.A.); nallatt.ocarez@gmail.com (N.O.); francolopezco@gmail.com (F.L.); 2Genetics & Biotechnology Lab, Plant & AgriBiosciences Research Centre (PABC), Ryan Institute, University of Galway, University Road, H91 REW4 Galway, Ireland; 3Instituto de Investigaciones Agropecuarias (INIA), La Platina, Research Centre, Av. Santa Rosa 11610, Santiago 8820000, Chile

**Keywords:** carotenoids, Lycopene β-cyclase, carrot, salt stress tolerance, *Actinidia deliciosa*

## Abstract

Plant carotenoids are synthesized and accumulated in plastids through a highly regulated pathway. Lycopene β-cyclase (LCYB) is a key enzyme involved directly in the synthesis of α-carotene and β-carotene through the cyclization of trans-lycopene. *Daucus carota* harbors two *LCYB* genes, of which *DcLCYB2* (annotated as *CCS-Like*) is mostly expressed in mature storage roots, an organ that accumulates high α-carotene and β-carotene content. In this work, we determined that *DcLCYB2* of the orange Nantes variety presents plastid localization and encodes for a functional LCYB enzyme determined by means of heterologous complementation in *Escherichia coli*. Also, ectopic expression of *DcLCYB2* in tobacco (*Nicotiana tabacum*) and kiwi (*Actinidia deliciosa*) plants increases total carotenoid content showing its functional role in plants. In addition, transgenic tobacco T2 homozygous plants showed better performance under chronic salt treatment, while kiwi transgenic calli also presented a higher survival rate under salt treatments than control calli. Our results allow us to propose *DcLCYB2* as a prime candidate to engineer carotenoid biofortified crops as well as crops resilient to saline environments.

## 1. Introduction

Carotenoids are colored isoprenoid pigments derived from the secondary metabolism in plants and algae but are also produced in some yeast and bacteria [1,2]. In plants, these pigments are involved in several significant processes, such as light absorption during photosynthesis [3,4], photoprotection via energy dissipation [5], and reactive oxygen species (ROS) detoxification [6,7]. In addition, they act as precursors of hormones such as abscisic acid (ABA) and strigolactones [8,9,10,11]. Carotenoids also have significant health benefits for humans, being precursors for vitamin A, acting as antioxidants, and preventing age-related diseases [12,13].

In plants, carotenoids are mainly synthesized in plastids of photosynthetic (leaves) and non-photosynthetic organs (fruits, flowers, and some roots) [14]. Carotenoid biosynthesis is a highly regulated process that involves multiple enzymatic steps [15,16,17] (Figure 1). The first enzyme in this pathway is phytoene synthase (PSY), which condenses two molecules of Geranylgeranyl diphosphate (GGPP) in the first non-colored carotenoid phytoene [14,18,19]. Subsequently, phytoene is desaturated and isomerized by the action of phytoene desaturase (PDS), 15-cis-f–carotene isomerase (ZISO), ζ-carotene desaturase (ZDS), and carotenoid (pro-lycopene) isomerase (CRTISO) to form the red-colored all-*trans* lycopene [20,21,22]. After lycopene, the pathway is divided into two branches; in the first branch, the orange-colored α-carotene is synthesized by the joint action of lycopene ε-cyclase (LCYE), producing the δ-carotene intermediate and lycopene β-cyclase (LCYB) [23]. Afterward, α-carotene can be hydroxylated by a β-ring hydroxylase (CHYB) to form zeinoxanthin, which is then hydroxylated by an ε-ring hydroxylase (CHYE) to produce lutein, a yellow-colored pigment [24,25]. On the second branch, the orange-colored β-carotene is produced by the cyclization of both ends of the all-*trans* lycopene catalyzed only by LCYB, producing γ-carotene as an intermediate molecule [23,26,27]. It is important to note that LCYB is required for both β-carotene and α-carotene synthesis. Then, by the double action of CHYB, β-carotene is converted first into β-cryptoxanthin and then into zeaxanthin, both yellow-colored pigments. This second β-branch (the β-carotene branch) is the one producing the metabolic precursors for hormones such as strigolactones and ABA [11,28]. Although the biosynthesis of carotenoids has been extensively studied in plant models such as *Zea mays* (maize) and *Arabidopsis thaliana* [29,30], the understanding of carotenoid synthesis in roots is still limited. Thus, carrot has emerged as a valuable plant model, because it synthesizes and accumulates large amounts of α-carotene and β-carotene specifically in the storage roots [31,32,33,34,35,36], suggesting an important role of LCYB in carrot storage root. Just et al., 2007 [37] identified the complete cDNA sequences for most of the carotenogenic genes of *D. carota.* Among them, in the *Daucus carota* V2.0 genome (Phythosome database), one sequence for *LCYB* named *DcLCYB1* (DQ192190) and *DcLCYB2* classified as capsanthin-capsorubin synthase (*CCS*; DQ192191, DCAR_022896) were identified [38]. Likewise, in the carrot genome uploaded to NCBI (ASM162521v1), the *DcLCYB2* sequence is also annotated as *CCS-like* (LOC108227824). Because carrots have never been found to accumulate or produce either capsanthin or capsorubin, it was proposed that it is more likely that this *CCS* is instead an *LCYB* [37]. It has been proven by heterologous complementation in bacteria that the *DcLCYB2* (*CCS-like*; LOC108227824) of the orange-colored GD2003 carrot variety encodes for a functional LCYB [39]. Expression studies suggest that *DcLCYB2* transcription increases through the carrot taproot development along with the increase in the carotenoid levels [32,35,40], and that this expression is inhibited in roots exposed to light and modulated through development root states [32,41]. Indeed, it has been shown that *DcLCYB2* is highly transcribed in the root [35,41], while *DcLCYB1* seems to be highly expressed in photosynthetic tissue [35,42,43]. Recently, it was found that the absence of β-carotene in the storage root of a red Chinese carrot variety (D904) was due to a Serine insertion at position 140 that inactivates the catalytic activity of DcLCYB2 [39]. This information suggests an essential role for DcLCYB2 in carrot storage roots.

On the other hand, salt stress provokes negative impacts on plants triggering severe reductions in growth and yield. In carrots, salt produces osmotic and ionic stress leading to biomass and growth reduction [44]. Also, carrot storage root volume and petiole length significantly decreased, reducing the storage root diameter by 30% and leaf length by 12% when exposed to 300 mM of NaCl [45]. Salt stress induces the production of ABA, being necessary to also increase the production of these carotenoid-derived precursors in roots [46]. It has been reported that *DcPSY1* and *DcPSY2* are transcriptionally induced under salt treatment in the Nantes orange carrot var and in the “BHJS” red carrot var [45,47], and DcPSY2 overexpression in Nicotiana tabacum induces salt stress tolerance [18]. Moreover, *DcLCYB* expression is also induced in the “BHJS” red carrot variety [17,45].

Interestingly, *LCYB* genes also play a key role in conferring tolerance to abiotic stress conditions, such as drought and salinity [48,49,50]. In *Salicornia europaea*, a highly salt-tolerant plant, the inhibition of SeLCYB activity produces a reduction in salt tolerance, while the ectopic expression of *SeLCYB* in tobacco conferred increased salt tolerance [48]. Similarly, ectopic expression of other *LCYBs* in tobacco, such as *AtLCYB* and *DcLCYB1*, also increases carotenoid content and salt tolerance and reduces oxidative damage [48,49,50]. Also, the overexpression of *LCYB2* of sweet potato (*IbLCYB2*) produced a 1.3- to 2.7-fold increase in total carotenoid content and enhanced tolerance to salinity and drought stress [51]. This highlights the potential of *LCYB* genes as outstanding tools for crop improvement, not only focused on carotenoid biofortification but also on increasing yield and abiotic stress tolerance [50].

In this study, we aimed to functionally characterize the *Daucus carota* cv. Nantaise *DcLCYB2* and evaluate its potential as a biotechnological tool for increasing carotenoid content and improving tolerance to salt stress in model and crop plants. By means of heterologous complementation in *Escherichia coli* and quantitative reverse transcription polymerase chain reaction (qRT-PCR) in carrot, we determined that *DcLCYB2* encodes for an enzyme with LCYB activity that is preferentially expressed in the carrot taproot. Additionally, through ectopic expression of *DcLCYB2 Nicotiana tabacum* and the economically relevant *Actinidia deliciosa* (Kiwi plant), which is known to be susceptible to drought and salinity [52,53], we observed not only an increase in carotenoid content but also an enhanced tolerance to salt stress in both plants.

## 2. Results

### 2.1. Carrot DcLCYB2 Encodes for a Functional LCYB Enzyme with Plastid Localization

Amino acidic alignment of the DcLCYB1 (NP_001316108) and DcLCYB2 (NP_001316108) indicates that DcLCYB2 has conserved motifs related to lycopene B-cyclases such as the conserved β-LCY region, a dinucleotide binding site, a charged region, and the β-cyclase catalytic motif [23,54] required for lycopene β-cyclase activity (Figure 2A) that were also present in the characterized DcLCYB1 [43]. DcLCYB2 has a 52% aminoacidic identity, with DcLCYB1 having a pairwise positive identity of 69.5% (Figure 2A). Most differences can be found within the first 50 residues at the N-terminus, where the predicted chloroplast target peptide is located [43]. We could not detect a clear chloroplast targeting sequence in either of the two sequences using the TargetP-2.0 [55] and the Localizer (V1.0.4) [56] software. In order to determine the sub-cellular localization of the enzyme, the DcLCYB2:GFP fusion protein was transiently expressed in 2-month-old tobacco leaves. Using confocal laser microscopy, we observed a clear colocalization of the fusion protein with the chloroplasts autofluorescence (Figure 2B and Appendix A), as was observed in previous research for the DcLCYB1 protein [43], indicating that DcLCYB2 presents the expected plastid localization.

To determine the enzymatic functionality of DcLYCB2, an in vivo analysis through heterologous complementation of the *E. coli* BL 21 gold strain carrying a lycopene producer biosynthetic plasmid (pDS1B ΔcrtY) [57] was co-transformed with pET-Blue1/DcLCYB2 and pET-Blue1 (negative control) for complementation experiments. Carotenoids were extracted after complementation from bacteria liquid culture and analyzed by HPLC-RP. The mutant strain co-transformed with the empty cloning vector pET-Blue1 produced only lycopene, shown as a single peak in the chromatogram at 474 nm (Figure 2C), while the mutant strain transformed with pET-Blue1/DcLCYB2 plasmid presented three peaks, two of which corresponds to lycopene and β-carotene, and a third peak that corresponds to γ-carotene (Figure 2C and Appendix A), a monocyclic intermediary in the conversion of lycopene to β-carotene. This points out that *DcLCYB2* of the orange carrot var Nantes encodes for a functional enzyme with lycopene beta-cyclase activity.

### 2.2. DcLCYB2 Is Preferentially Expressed in the Carrot Taproot and Induced by Salt Treatment

In addition, a phylogenetic tree based on 19 *LCYB* nucleotide sequences of different plant species (Appendix A) placed the *DcLCYB2* sequence within the “fruit expression clade”, where other *LCYB2* (including *Actinia chinensis* LCYB1 and LCYB2) were grouped separated from the *LCYB1* clade where they are classified as of “Photosynthetic organ expression”. *DcLCYB2* is more related to the tomato *CYCB* and pepper *CCS*, which codifies for CYCB and CCS, respectively; both enzymes are expressed in fruits and transform lycopene into β-carotene [54,58]. This suggests a differential organ function for both *DcLCYB1* and *DcLCYB2* genes. In fact, expression analysis by qRT-PCR (Figure 2D) showed that *DcLCYB1* and *DcLCYB2* are transcribed in roots and leaves in 4- and 12-week-old plants, but *DcLCYB1* is preferentially expressed in leaves, while *DcLCYB2* is mainly expressed in mature storage roots (12-week-old plants). This suggests that although both *DcLCYBs* are expressed in leaves and storage roots, *DcLCYB2* predominates in the mature taproot, as previously reported [35,41].

To evaluate if the expression of *DcLCYBs* is induced in the storage root of the Nantes orange carrot var. under salt stress, 8-week-old carrot plants were exposed to acute 250 mM NaCl treatment for 2–8 h. *DcLCYB1* and *DcLCYB2* transcript levels presented a significant increase displaying a clear accumulation pattern in 250 mM NaCl treatment at 4 h and 8 h of treatment (Figure 2E). 

### 2.3. Ectopic Expression of DcLCYB2 in N. tabacum Boosts Carotenoid Content and Increases Tolerance to Salt Stress

Considering the previous finding, we propose evaluating the participation of *DcLCYB2* in the induction of the carotenogenic biosynthetic pathway and in conferring salt stress tolerance. For this, the coding sequence of *DcLCYB2* was cloned under the direction of the 35S promoter in the binary vector pMDC32. Plants were transformed and selected by Hygromycin resistance, and several lines were subjected to RT-PCR (Appendix A), selecting L2, L3, L4, L7, and L9 for pigment quantification. We found that all selected transgenic lines showed an increase in the total carotenoid content, mainly in the β-carotene and lutein levels along with chlorophyll levels (Figure 3A) which correlate with an increase in the transcript levels of endogenous *NtPSY1* and *NtPSY2* and *DcLCYB2* transcript levels (Appendix A). These findings reflect that the ectopic expression of *DcLCYB2* in tobacco increments the metabolic flux of both carotenoid biosynthesis branches as well as the common metabolic precursors shared with chlorophylls [16,59].

To further inquire about the effect on salt stress tolerance in DcLYCB2 transgenic tobacco lines, two-month-old Hygromycin-resistant T_1_ from L4 and L9 (lines with higher carotenoid levels) were subjected to chronic salt treatment for 4 weeks and then irrigated with water for a week for recovery. This treatment was repeated one more time (a total of 10 weeks of treatment), and then lines were phenotypically analyzed. T1L4 and T1L9 transgenic lines showed greener leaves by the end of the treatment, displaying fewer chlorotic leaves and wilting when compared to the wild-type (WT) and EV tobacco plants (Figure 3B). In addition, we analyzed the effect of chronic salt treatments on root elongation, total carotenoids, and germination rate of homozygous T_2_ transgenic tobacco seedlings. One-week-old seedlings were exposed to 200 mM NaCl treatment. After 14 days, roots of homozygous transgenic T_2_ lines (L4.10, L4.17, L9.7, and L9.10) presented longer principal roots in the control condition but significantly longer principal root length under salt treatment when compared to the WT (Figure 3C). Also, under salt conditions, all transgenic lines analyzed exhibited a significative increment in carotenoid and chlorophyll levels compared with WT tobacco seedlings under the same conditions (Figure 3D). Therefore, we decided to analyze the survival rate by germinating 50 seeds of each line in MS semisolid plates, supplemented with 200 mM NaCl and control conditions (without NaCl). After 14 days, WT and all transgenic lines germinated within the first 3 days in control conditions (Figure 3E), achieving germination over 80% by the end of the first week. Surprisingly, some of the transgenic lines showed a slight delay and lower germination rate (90%) when compared to WT (100%) at the end of the assay. On the other hand, transgenic salt-treated seeds started to germinate 6–7 days after sowing, while WT seeds hardly germinated after 10 days. Specifically, the transgenic L9.10 line exceeded a 50% germination rate by day 10, and the other transgenic lines (L4.10, L4.17, and L9.7) surpassed 50% by day 13. At the end of the experiment (day 14), only 6% of the WT seeds germinated; all transgenic lines showed a germination rate higher than 65%, with lines L9.7 and L9.10 achieving an almost 90% germination rate (Figure 3E). These results highlight that *DcLCYB2* enhances carotenoid and chlorophyll synthesis, maintains the greenness of the leaves in transgenic plants, as well as promotes greater carotenoid production, longer principal root length, and germination rate under salt stress.

### 2.4. Ectopic Expression of DcLCYB2 in A. deliciosa Increase Salt Tolerance in Transgenic Callus and Boost Carotenoid Content in Leaves

*A. deliciosa* is a salt-sensitive cultivar that leads to decreased vegetative growth, reduced vegetative development, and lower photosynthetic yield when exposed to saline conditions [52,53]. The development of transgenic kiwi plants with increased carotenoids and tolerance to salt stress has been proposed as a useful strategy to overcome this issue [60,61]. Of note, it has been reported that *LCYB* plays a key role in carotenoid biosynthesis and accumulation in kiwi [62]. By following the previously published transformation protocol [61], we obtained *A. deliciosa*-transformed calli that were subjected to chronic NaCl survival assay [62] to preliminarily evaluate the salt tolerance conferred by the ectopic expression of *DcLCYB2*. Two-month-old transformed calli were placed on a solid half-strength MS medium supplemented with ascending NaCl concentrations, and survival rate was determined at 5 and 15 days of chronic treatment (Figure 4A,B and Appendix A) by quantification of green living calli or dark dead calli. By the first measurement on day 5 of NaCl screening, neither the WT nor the EV lines survived the 200 mM NaCl while the *DcLYCB2*-transformed calli showed a 24% of green living calli (Appendix A). At the end of the experiment on the 15th day, 40% of the *DcLCYB2*-transformed calli survived at 100 mM of NaCl, but only 25% of WT plants did, while only *DcLCYB2*-transformed calli (8%) survived on the highest concentration of NaCl (150 mM and 200 mM of NaCl, Figure 4A,B). Additionally, transformed calli that survived on the 15th day were recovered and further analyzed by RT-PCR, confirming that they are expressing *DcLCYB2* (Appendix A). Then, we regenerated eight transformant lines, of which four were confirmed to express the *DcLCYB2* transgene (Figure 4C). We found that three of the transgenic lines (L3, L4, and L8) had a significant increase in β-carotene and total carotenoid in their leaves (Figure 4D), while L3 and L4 also present an increment in total carotenoid and lutein content. Controversially, L6 did not present any difference in β-carotene and total carotenoid content but a significant reduction in lutein (Figure 4D). Even so, if we consider L3 and L4, which presented the most consistent results, we can suggest that the *DcLCYB2* ectopic expression in kiwi, as well as that shown in tobacco, participates in both carotenoid biosynthesis branches by increasing the β-carotene and lutein content. Altogether, these results indicate that the expression of *DcLYCB2* produced salt-tolerant kiwi calli and increased the carotenoid content in this sensitive cultivar.

## 3. Discussion

### 3.1. Carrot DcLCYB2 Encodes a Functional LCYB Enzyme

A partial pyridine nucleotide binding site is a conserved domain located in the active center of the lycopene cyclase enzymes [23,63,64]. The signal peptide in LCYB is found in the first 50–100 amino acids [54,65], and a conserved motif called the “conserved β-cyclase region” is found within the signal peptide that is proposed as a main factor for membrane association [54] and as an essential requirement for a correct catalytic activity [27]. In LCYBs of plants and bacteria, a conserved “dinucleotide binding motif” is also found [66]. In addition, several conserved domains, such as the cyclase motifs I and II and a charged region, are highly conserved in plants that were also found in DcLCYB1 and DcLCYB2 (Figure 2A). The “β-cyclase motif” domain is fully conserved in all plants’ LCYBs. These motifs and regions could be involved in the substrate–enzyme interaction and in the catalysis [23,54,67].

Phylogenetic analyses suggest a common origin of the *LCYB*, *CCS*, and *Neoxanthin synthase* (*NSY*) genes [65,68], which code for enzymes with a very similar reaction mechanism [67,68,69]. It has been reported that the *LCY* gene family from plants, algae, and bacteria share a common ancestor, which gave rise to the Ε-ring and β-ring. Within the LCYB group, the CCS/NSY include plant enzymes that function as capsanthin-capsorubin synthases (CCS) and/or neoxanthin synthases (NSY) but, in some cases at least, retain the ability to function as a lycopene-ring cyclase [70]. Indeed, LCYB catalyzes a simplified version of the reactions catalyzed by NSY and CCS, suggesting that these two enzymes could have originated from LCYB during the evolution of higher plants to give rise to new oxygenated carotenoids [71]. Other evidence indicating that NSY and CCS originated from a gene duplication of LCYB is that in cyanobacteria, the supposed progenitors of plastids, there are no NSY nor CCS, but there is LCYB [65,69]. Specifically, this is the case in the phylogenetic tree that we performed, which grouped DcLCYB2 within the “fruit” clade and was related to CCS-like enzymes (Appendix A). It would be interesting to further determine if DcLCYB2 is a bifunctional enzyme, as reported in paprika [67]. But it is also important to consider that carrot does not accumulate either capsanthin or capsorubin [37].

β-Carotene accumulation has been widely examined in a variety of crops and model species, like Arabidopsis [30], maize [72], and tomato [45]. Despite this, the genetic regulation of β-carotene accumulation in carrot, which is one of the most abundant sources of this nutrient, is still incomplete. In several plant models such as *Arabidopsis thaliana* [73], *Oryza sativa* [74], and *Zea mays* [68,75], the LCYB enzyme is encoded by only one gene, whereas in orange carrots that accumulate high levels of carotenoids in non-photosynthetic organs, two or more isoforms for *PSY*, *ZDS*, *LCYB,* and *NCED* have been reported [33,37,76]. In these cases, both genes tend to be differentially expressed between photosynthetic and non-photosynthetic organs. In tomato, the *LCYB* gene shows a preferential expression in green organs, while the *LCYCB* gene presents an organ-specific expression and function in ripening fruits and flowers [58,77,78]. Similar organ-specific functions of two *LCYB* genes have also been described in *Capsicum annuum* [54], *Carica papaya* [79,80,81], *Crocus sativus* [82], and *Citrullus lanatus* [26]. In our model, we also found tissue-specific expression patterns between *DcLCYB1* and *DcLCYB2* (Figure 2D). *DcLCYB1* displayed a steady expression pattern throughout root development; meanwhile, *DcLCYB2* showed an increased expression in the storage root at week 12 (Figure 2D). At the functional level, DcLCYB1 and DcLCYB2 present plastid subcellular localization (Figure 2B and Appendix A [43]) and β-cyclase activity (Figure 2C [43]), which were comparable with the findings reported in the orange carrot GD2003 variety [39], showing a conserved functionality among orange cultivars. Indeed, *DcLCYB1* silencing produces a 70% reduction in total carotenoid and β-carotene levels in leaves and a 55% reduction in the storage root [43]. This evidence led to the proposal that another LCYB, probably DcLCYB2, is also required for total carotenoid and β-carotene synthesis in the orange carrot storage root. Considering that the S140L mutation in the red D904 carrot variety inactivates the catalytic activity of DcLCYB2, it can be concluded that this mutation is responsible for the low β-carotene and high lycopene accumulation in this variety [39]. Therefore, DcLCYB2 presents an essential role in carotenoid synthesis in carrot storage roots.

### 3.2. DcLCYB2 as a Biotechnological Tool to Boost Carotenoid Content and Increase Tolerance to Salt Stress in Plants

Interestingly, either *DcLCYB1* or *DcLCYB2* is transcriptionally induced in carrot storage roots of plants submitted to acute salt treatment, suggesting a role of both LCYBs in the response to salt stress (Figure 2E). It has been shown that chemical inhibition of lycopene cyclization was reported to produce salt sensitivity under sub-optimal salt concentrations in *Salicornia europea*, reflecting the important role of lycopene cyclization and the downstream carotenoid biosynthetic pathway in salt stress tolerance. Transgenic tobacco and Arabidopsis expressing *SeLCYB* exhibited enhanced salt tolerance by alleviation of oxidative damage through enhanced carotenoid production [48]. Previously, RNAi silencing of *NtLCYB* was reported to lead to a decrease in biomass, shorter main roots, and earlier germination due to a reduction in the GA/ABA content with the concomitant decrease in β-carotene [83]. The ectopic expression of *DcLCYB1 in N. tabacum* also induces the transcription of endogenous carotenogenic genes *NtPSY1* and *NtPSY2,* resulting in an increment of total carotenoids and chlorophyll levels [43] but also an increase in plant fitness overall, rendering greater biomass with higher gibberellin levels [59,84]. In concordance, ectopic expression of *DcLCYB2* in tobacco plants also resulted in increased transcription of *NtPSY1* and *NtPSY2* endogenous carotenogenic genes (Appendix A) accompanied by an increase in total carotenoid content, mainly β-carotene (Figure 3A).

*DcLCYB1* and *DcLCYB2* transcript levels presented a significant increase displaying a clear accumulation pattern in 250 mM NaCl at 4 h and 8 h of treatment (Figure 2E). This result agrees with the findings of Zhao et al., 2022 where the *DcLCYB* expression is induced in the “BHJS” red carrot variety stressed with 300 mM of NaCl after 80 days after sowing [73]. DcLCYB2 transgenic tobacco lines subjected to salt treatment showed better performance, with reduced chlorosis in leaves (Figure 3B), increased principal root length (Figure 3C), higher carotenoid and chlorophyll content (Figure 3D), and improved seed germination (Figure 3E) when compared with control plants. Surprisingly, even when the germination rate of the DcLCYB2 transgenic lines is greater in saline conditions, under control conditions, the transgenic lines showed a slightly lower and delayed germination, reaching, in most cases, a germination rate of around 90%. As β-carotene is a metabolic precursor of ABA, which delays and inhibits germination [85,86], we can suggest that the transgenic tobacco seeds produce an increase in ABA due to the redirection or enhancement of the metabolic flux in the β-branch by the effect of *DcLCYB2* ectopic expression.

Likewise, the effect of *DcLCYB2* can be transferred to economically important crops, like *Actinidia deliciosa* (kiwi), producing a significant increase in the total carotenoid and β-carotene content in leaves (Figure 4C), considering that transgenic kiwi calli expressing *DcLCYB2* showed an improved survival rate under chronic salt treatment (Figure 4A,B and Appendix A). Altogether, these results, in addition to the recent publications in carotenoids and abiotic stress [50,84,87], allow us to propose *DcLCYB2* as a prime candidate to be engineered with the dual purpose of generating biofortified crops as well as crops resilient to the effects of climate change.

## 4. Materials and Methods

### 4.1. Plant Material

Seeds of commercially acquired carrot (*Daucus carota* cv. Nantaise improved 3) were grown under controlled climate conditions in a greenhouse located inside the Faculty of Science at the University of Chile with a 16 h photoperiod, illuminated with white fluorescent light (150 µmol m^−2^ s^−1^) at 20–23 °C. Wild-type (WT) and transgenic tobacco (*Nicotiana tabacum*) seeds were surface sterilized in a solution containing 95% ethanol for 1 min, then in sodium hypochlorite 30% (*v*/*v*) plus 0.01 tween-20 for 25 min, and finally washed with sterile water and dried on sterile paper. The seeds were then germinated in vitro in solid MS plates (4.4 g/L MS salts, 1% sucrose, 0.01% Myo-inositol, 0.22% vitamins, and 0.7% agar pH 5.7 with or without antibiotic) and maintained at 22 °C in a growth chamber with a 16 h long day photoperiod (white fluorescent light; 115 μmol m^−2^ s^−1^). WT or transgenic lines were maintained in vitro and then transferred to a greenhouse and cultivated in soil/vermiculite (2:1) for all further analyses. Kiwi plants (*Actinidia deliciosa* cv. Hayward, CA, USA) were a generous gift from Viverosur Ltd.a (Longitudinal Sur Km 174, Teno 174, Curicó, Chile) and maintained in vitro, as previously reported [61].

### 4.2. Vector Construction

Carrot cDNA was obtained with Impron II reverse transcriptase (Promega^®^, Madison, WI, USA). DcLcyb2_F and DcLcyb2_R or DcLcyb2-nst (Appendix A) primers were used for RT-PCR amplification of *DcLCYB2* with or without stop codon (1612 and 1609 bp, respectively) with Pfu DNA polymerase (Fermentas). Amplified fragments were cloned into the entry vector, pCR^®^8/GW/TOPO (pCR8; Invitrogen), following the manufacturer’s instructions. Positive clones were verified by enzymatic digestion and sequencing (Macrogen Corp., Rockville, MD, USA). Afterward, pCR8/*DcLCYB2* was recombined in the pMDC32 plasmid [88] to generate the pMDC32/*DcLCYB2* plasmid by using the Gateway^TM^ LR-clonase II enzyme mix (Thermofisher, Waltham, MA, USA) according to the manufacturer’s protocol. Similarly, the pCR8/*DcLCYB2*_ns (no stop) was recombined into pGWBB5 plasmid [89], obtaining the binary vector pGWB5/DcLCYB2 that will produce the fusion DcLCYB2:GFP protein. For heterologous complementation, *DcLCYB2* was amplified with DcLcyb2_F and DcLcyb2-nst primers (Appendix A) from pCR8/*DcLCYB2* plasmidial DNA, purified and cloned in the EcoRV site of the pET-Blue1 (NovaBlue^®^, Dubai, United Arab Emirates) expression vector. Positive clones that harbor the gene in the sense orientation with respect to the T7 promoter were selected by enzymatic digestion and sequencing, creating the pET-Blue1/*LCYB2* bacterial vector.

### 4.3. Heterologous Complementation in Escherichia coli

Functional assays were carried out in *Escherichia coli* BL 21 gold strain as described in Moreno and Stange (2022) [90]. *E. coli* BL21 cells were transformed with the pDS1B or pDS1BΔcrtY plasmid (designed and gifted by Dr Victor Cifuentes, Biotechnology Center, Facultad de Ciencias, Universidad de Chile). The pDS1B carries the carotenogenic genes of *Erwinia uredovora* to produce β-carotene. The pDS1B∆crtY vector has a mutation in the crtY gene that codifies for lycopene β-cyclase, leading to the accumulation of lycopene (Figure 2). This mutant strain was transformed with the pET-Blue1/*LCYB2*. The empty pET-Blue1 vector was included as control. The transformed colonies were selected in LB medium supplemented with ampicillin (100 µg mL^−1^) and chloramphenicol (50 µg mL^−1^) and incubated for 96 h at 30 °C. A preculture of pET-Blue1/*LCYB2* and pET-Blue1 plasmids was used to inoculate 200 mL of LB medium with the selective antibiotics. When the culture reached O.D600:0.6, 100 µL of IPTG 1M (isopropyl β-D-thiogalactoside) was added to half of the culture to induce the expression of the gene, and the other half was used as negative control. All the assays were performed in triplicate and in darkness (to maximize carotenoid production).

### 4.4. Plant Transformation

Transformation of *N. tabacum* with the pMDC32/*DcLCYB2* or pMDC32 (EV) plasmid was undertaken by Agrobacterium-mediated transformation using the bacterial strain GV3101::pMP90 as reported previously [8,91]. For the selection of transgenic T0, T1, and T2 lines, Hygromycin (10 μg ml^−1^) was used. *A. deliciosa* was transformed with Agrobacterium harboring the pMDC32/*DcLCYB2* or EV plasmid [61]. Transformed plants were selected with increasing Hygromycin concentration through the successive subcultures (5 to 20 μg ml^−1^).

### 4.5. Acute and Chronic Salt Treatments

For chronic salt treatments, 8-week-old T_1_
*N. tabacum* lines were irrigated twice per week with 250 mM NaCl (100 mL of each salt solution per plant individually cultivated in 300 cm^3^ pot) in 1/8 MS liquid media for 4 weeks; then, 1 week of recovery occurred by watering without NaCl, and then the process was repeated one more time, reaching 10 weeks. In the end, the number of green and yellow leaves was quantified, and images were captured to compare the phenotypes of the plants. For this experiment, 28 WT plants and 8–13 transgenic lines (EV, L4, and L9) were used. Germination rate was carried out with 50 seeds of each T_2_ line cultivated in MS media with vitamins and 1% sucrose (control) or supplemented with 200 mM NaCl for 14 days in a 16 h photoperiod at 20–23 °C in a growing chamber. Root elongation, carotenoid, and chlorophyll quantification were carried out in these two-week-old T_2_ seedlings grown in MS or MS + NaCl in three pools of three plants each. For *A. deliciosa* calli salt treatment, 2-month-old kiwi WT or transformed with *DcLCYB2* and EV calli [61] were transferred to a callus proliferation media supplemented with 0, 50, 75, 100, 150, or 200 mM NaCl. At 5 and 15 days post-treatment, the calli survival rate was quantified, and images were taken. The experiment was performed in triplicate with 7 calli per replicate and performed independently two times. For carrot acute salt treatment, 8-week-old WT plants were immersed for 2, 4, and 8 h in MS supplemented with 250 mM NaCl. After each time, carrot root was sampled. The assay was performed with 3 biological replicates for each sampling time (3 plants per sample), each with 2 technical replicates. Then, total RNA was isolated for gene expression analysis.

### 4.6. Carotenoid Extraction and Spectrophotometric Quantification

Carotenoids from *E. coli* BL21 complemented cells were extracted from 70 mL of liquid cultures as described in Moreno and Stange (2022) [90]. The bacterial pellet was washed twice with cold sterile water and resuspended in 1 mL of sterile water with 500 µL of glass pearls. After vigorous mixing, 1 mL of acetone was added, mixed, and centrifuged at 4000 rpm for 5 min. The water/acetone solution was collected, and the pellet was washed twice with 1 mL acetone (or until a white pellet was obtained). All supernatants were mixed and dried; the resulting dried sample was mixed with petroleum ether in a proportion of 1 (aqueous phase):4 (petroleum ether). This solution was centrifuged at 4000 rpm for 5 min, and the upper phase was collected in glass tubes, dried with gaseous nitrogen, and resuspended in 100 µL of acetone for Spectrophotometric and HPLC-RP analysis. For plant carotenoid extraction [43], 100 mg of T_0_ tobacco or T_0_ kiwi leaves were ground with 1 mL of hexane/acetone/ethanol (2:1:1 *v*/*v*). Two successive extractions were performed until the tissue was blanched to extract all carotenoids. The extract was evaporated with gaseous nitrogen and resuspended in 2 mL of acetone for Spectrophotometric and HPLC-RP analysis. Total carotenoids were measured by spectrophotometry at 750, 662, 645, and 470 nm in quartz cuvettes. Absorbance at 662, 645, and 470 nm is used to determine the concentration of chlorophyll a, chlorophyll b, and total carotenoids, respectively [92], and the absorbance at 750 nm was measured to determine the turbidity of the sample.

### 4.7. High-Performance Liquid Chromatography in Reverse Phase (HPLC-RP)

The measurements were made in Shimadzu HPLC equipment (LC-10AT) with a diode array, and the data analysis was carried out with the LCsolutions^®^ (Doral, FL, USA) software program. The pigments were separated by an HPLC using an RP-18 Lichrocart 125-4 reverse phase column (Merck^®^, Rahway, NJ, USA), utilizing acetonitrile/methanol/isopropanol (85:10:5 *v*/*v*) as a mobile phase with a 1 mL/min flow at room temperature in isocratic conditions. The elution specters of each maximum were obtained using a diode array detector. Carotenoids were identified according to their absorption spectra, time retention, and comparison with pigment-specific standards, which was corroborated by comparison with the *Carotenoids: Handbook* [1]. All assays were carried out in triplicate, on ice, and in dark conditions.

### 4.8. RNA Extraction and qRT-PCR

Total RNA was extracted from frozen powder of 200 mg *Daucus carota* L cv. Nantes leaves and storage roots of 4-, 8- and 12-week-old plants and from two-month-old T_0_ DcLCYB2 tobacco transgenic plants using RNAsolv (Omega Biotec, GA, USA). Genomic DNA traces were eliminated by a 20 min RNase-free DNase I treatment. For cDNA synthesis, approximately 2 μg of total DNA-free RNA was mixed with 1 mM of oligo dT primer and Impron II Reverse Transcriptase (Promega^®^, Madison, WI, USA). Real-time RT-PCR experiments were performed in a Stratagene Mx3000P thermocycler, using SYBR Green double-strand DNA binding dye, as described in Stange et al., 2008 [31]. Specific primers targeting *LCYB2* based on published cDNA sequences and the reference [37,38] (Appendix A) were used. As normalizer gene, carrot *ubiquitin* (DCU68751) and tobacco *RNAr18S* (AJ236016.1) were used. Primer efficiencies were determined by amplification of the target from a PCR dilution series in triplicate using melt curve analysis according to the manufacturer’s instructions (Stratagene-SYBR Green catalog) and according to the equation E = (10 (−1/slope)) − 1. Final data were obtained by introducing fluorescence results in the equation described by Pfaffl (2001) [93]. Each qRT-PCR reaction was performed using three biological replicates, and each sample was analyzed twice (technical replicate). The reaction specificities were tested with melting gradient dissociation curves and electrophoresis gels in all cases.

### 4.9. Subcellular Localization

The construct pGWBB5/DcLCYB2:GFP was transiently expressed in 2-month-old *N*. *benthamiana* tobacco leaves by agroinfiltration, as reported previously [20]. Samples were visualized in a Zeiss LSM5 confocal microscope at 498–525 nm (GFP) and 640–720 nm (Chlorophyll) and processed with the ImageJ software.

### 4.10. Statistical Analysis

To test for significant differences in gene expression, all statistical analyses were carried out using the statistical software package Graphpad Prism.

## Figures and Tables

**Figure 1 plants-12-02788-f001:**
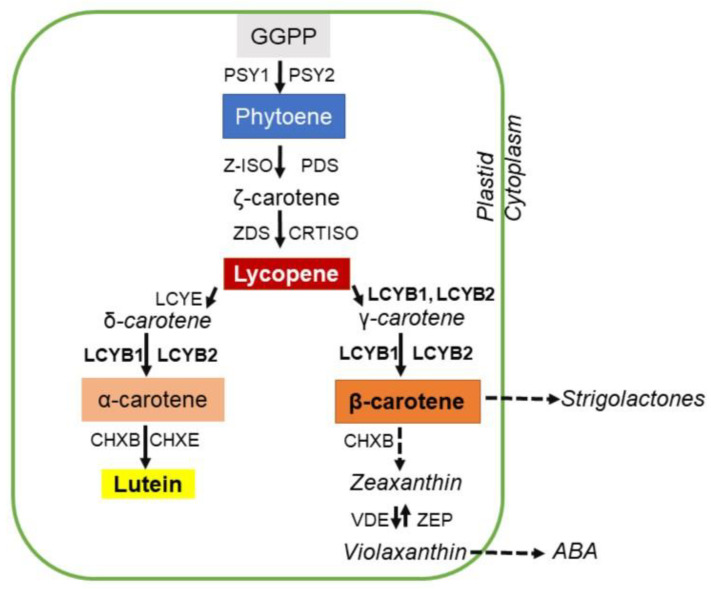
Illustrative diagram of the steps involved in the biosynthesis of carotenoids in plants. The process begins with geranylgeranyl diphosphate (GGPP), which is converted into phytoene by phytoene synthase (PSY1 and PSY2) and then into ζ-carotene by phytoene desaturase (PDS) and ζ-carotene desaturase (ZDS). Other enzymes involved in the pathway such as carotene isomerase (CRTISO), lycopene ε-cyclase (LCYE), lycopene β-cyclase (LCYB1 and LCYB2), β-carotene hydroxylase CHXB), ε-carotene hydroxylase (CHXE), zeaxanthin epoxidase (ZEP), and violaxanthin de-epoxidase (VDE) are required for colored carotenoid synthesis (in plastids) as well as for abscisic acid (ABA) and strigolactones (in the cytoplasm). In this study, carrot *LCYB2* gene was examined, along with its involvement in β-carotene accumulation and its participation in salt tolerance in heterologous plants.

**Figure 2 plants-12-02788-f002:**
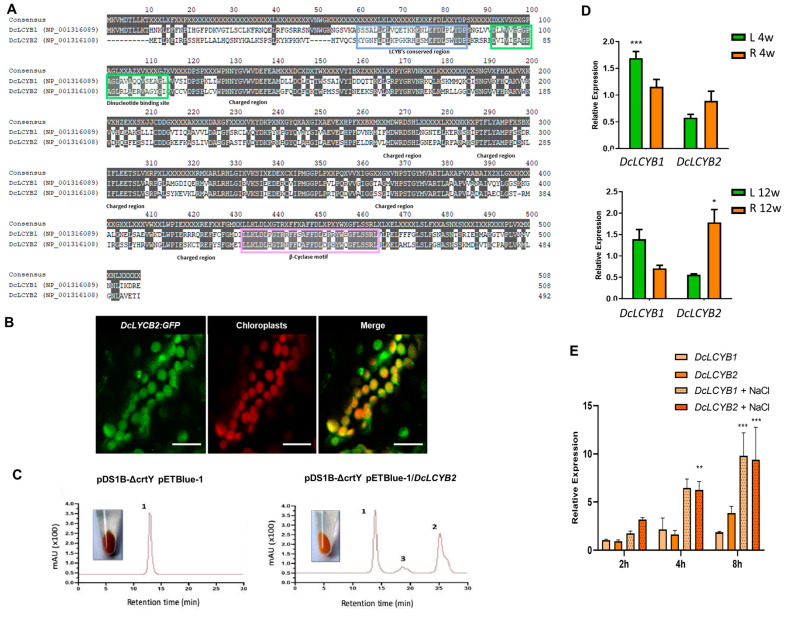
Bioinformatic analysis of DcLCYB2 and functional study through heterologous expression in *Ecoli* and subcellular localization. (**A**) Alignment of the deduced amino acid sequences of DcLCYB1 and DcLCYB2. The alignment was created using Geneious Prime software (v2023.0.1) with the Clustal Omega aligner. Numbers on the right denote the number of amino acidic residues. Identical amino acids are shown in white text and grey background. Different residues are denoted with an X on the consensus. Characteristic regions of plant β-LCYs are indicated in rectangles: β-LCY conserved region in cyan, dinucleotide binding site in green and β-Cyclase motif in purple, also conserved charged regions are noted [23,54]. (**B**) Subcellular localization of DcLYCB2. Representative of confocal laser images from transient transformation of 2-month-old tobacco leaves after 4 days with DcLYCB2:GFP (left), chloroplast autofluorescence (mid), and the merge (right) indicating a colocalization of DcLYCB2:GFP at the chloroplasts. Bar 10 µm. (**C**) RP-HPLC analysis of carotenoids extracted from *E. coli* cells complemented with *DcLCYB2*. Chromatograms of carotenoids extracted from pDS1B-ΔcrtY-transformed *E. coli* BL21 pETBlue-1 without insert (negative control) or pETBlue-1/DcLCYB2. The HPLC elution profiles of the control cultures containing the empty pETBlue-1 cloning vector show a single peak whose retention time and absorption spectrum correspond to lycopene. The chromatograms of the co-transformed strain with the pET-Blue1/DcLCYB2 plasmid show two new peaks corresponding to β-carotene and another carotenoid. The bacterial pellet of each transformed strain is shown to the left of each chromatogram. AU: absorbance units. (**D**) Relative expression of *DcLCYB1* and *DcLCYB2* in leaves (L) and storage roots (R) at 4 weeks (4 w) and 12 weeks (12 w) of development. The average Ct values of each gene of the R samples were used as calibrators. *DcUbi* was used as normalizer. Columns and bars represent the means and SE (n = 3). The asterisks show significant differences between the two analyzed genes that were determined using a non-paired *t*-test with Welch correction *** (*p* < 0.001), * (*p* < 0.01) (**E**) Relative expression of *DcLCYB1* and *DcLCYB2* in storage roots under salt stress. Twelve-week-old carrot plants were irrigated with 100 mM NaCl, and samples were taken at 2, 4, and 8 h of treatment. The expression level was determined by qRT PCR. *DcUbi* was used as normalizer. Columns and bars represent the means and SE (n = 3). Significant differences between the two analyzed genes were determined using a non-paired *t*-test with Welch correction. *** (*p* < 0.001), ** (*p* < 0.05).

**Figure 3 plants-12-02788-f003:**
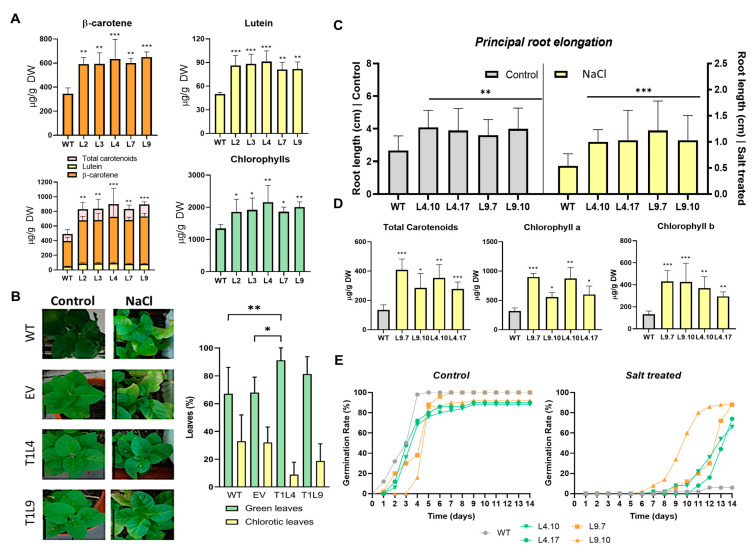
Effect of *DcLCYB2* ectopic expression in *N. tabacum*. (**A**) β-carotene, lutein, total carotenoids, and chlorophyll levels in WT and *DcLCYB2* transgenic T_0_ tobacco plants. (**B**) Qualitative and quantitative analysis of chlorotic leaves in *DcLCYB2* transgenic T_1_ and WT tobacco plants under chronic salt treatment (250 mM NaCl). (**C**) Principal root elongation of *DcLCYB2* transgenic T_2_ and WT tobacco plants under chronic salt treatment (200 mM NaCl). Every column corresponds to the mean of the dataset, which is comprised of 18 individual plants. Assays were subjected to a one-way ANOVA with Brown–Forsythe and Welch corrections comparing the mean of each dataset with the mean of the WT dataset. (**D**) Total carotenoid and chlorophyll content in *DcLCYB2* transgenic T_2_ and WT tobacco plants under chronic salt treatment (200 mM NaCl). (**E**) In vitro germination rate of *DcLCYB2* transgenic T_2_ and WT tobacco plants under control and chronic salt treatment (200 mM NaCl). All values represent the means of three independent replicates (+SD). For (**A**,**B**,**D**,**E**), statistically significant differences were determined by one-way ANOVA test and Bonferroni post-test: * *p* < 0.05, ** *p* < 0.01, and *** *p* < 0.001. Control: well-watered plants; WT: wild type.

**Figure 4 plants-12-02788-f004:**
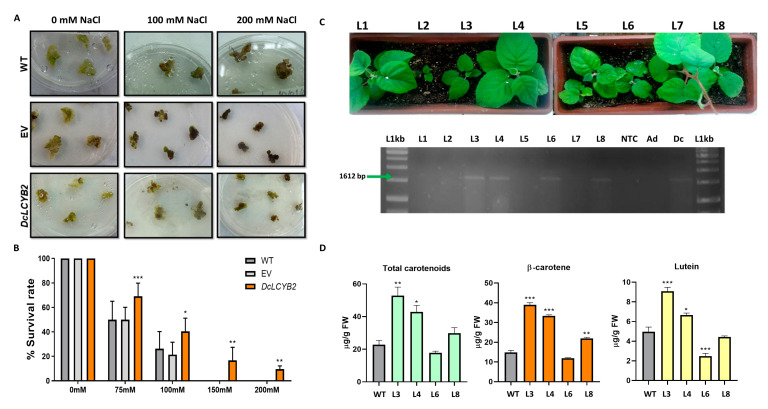
Effect of *DcLCYB2* ectopic expression in *A.deliciosa*. (**A**) Phenotype of *DcLCYB2* transformant and WT *A. deliciosa* calli after 15 days under chronic salt treatment (0 mM, 100 mM, and 200 mM NaCl). (**B**) Survival rate of *DcLCYB2* transformant and WT *A. deliciosa* calli after 15 days under chronic salt treatment with different concentrations of NaCl (0 mM, 75 mM, 100 mM, 150 mM, and 200 mM NaCl). The mean and SD of each bar represent the results from 6 plates (with 7 calli each) for WT, EV, or DcLCYB2 subjected to different NaCl concentrations. A two-way ANOVA test was performed, followed by a Dunnett’s multiple comparisons test to determine significant differences between DcLCYB2 and EV dataset against the WT control. (**C**) Phenotype and RT-PCR identification of *A. deliciosa* transgenic line expressing *DcLCYB2*. (**D**) Total carotenoids, β-carotene, and lutein levels in WT and *DcLCYB2* transgenic *A. deliciosa* leaves. Values of (**C**,**D**) represent the means of three independent replicates (+SD), and statistically significant differences were determined by one-tailed ANOVA test and Bonferroni post-test: * *p* < 0.05, ** *p* < 0.01, and *** *p* < 0.001. WT: wild type; EV: empty vector; NTC: No template control *DcLCYB2* RT-PCR control; Ad: wild-type *A. deliciosa*-negative *DcLCYB2* RT-PCR control; Dc: wild-type *D. carota*-positive *DcLCYB2* RT-PCR control; L1kb: 1kb ladder; bp: base pair.

## Data Availability

Data are contained within the article, and Appendix A is available at https://www.mdpi.com.

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
