# Peer review of "Putative Daucus carota Capsanthin-Capsorubin Synthase (DcCCS) Possesses Lycopene β-Cyclase Activity, Boosts Carotenoid Levels, and Increases Salt Tolerance in Heterologous Plants"

_plants, 2023, doi:10.3390/plants12152788_

Round 1

Reviewer 1 Report

I believe authors have done a good job showing a putative role of BLCY2 in carrot roots, together with detecting a potential role of carotenoids in enhancing salt resistance. I believe that this gene could be a great potential target for improving crop resilience to different stresses. Some minor comments are attached in pdf file in order to improve figures interpretation and statistics in graphs. 

Author Response

C1: I believe authors have done a good job showing a putative role of BLCY2 in carrot roots, together with detecting a potential role of carotenoids in enhancing salt resistance. I believe that this gene could be a great potential target for improving crop resilience to different stresses. Some minor comments are attached in pdf file in order to improve figures interpretation and statistics in graphs. 

R: We sincerely appreciate reviewer`s comment

Q1: After 14 days, roots of homozygous transgenic T2 lines (L4.10, L4.17, L9.7 and L9.10) presented longer principal roots in the control condition but significantly longer principal roots length under salt treatment when compared to WT (Figure 3C). Revise statistic and data to report real differences because it seems there are not significative differences in root length.

R1: Both assays were subjected to a One-way ANOVA with Brown-Forsythe and Welch corrections in order to compare the mean of each dataset with the mean of the WT dataset of the corresponding assay (Control or Salt treated). The statistical difference reported corresponds to the P value summary of Welch’s ANOVA test. Additionally, every column corresponds to the mean of the dataset, which is comprised of 18 individual plants.

Q2: Revise statistic and data to report real differences: B) Qualitative and quantitative analysis of chlorotic leaves in DcLCYB2 transgenic T1 and WT tobacco plants under chronic salt treatment (250mM NaCl), C) Principal root elongation of DcLCYB2 transgenic T2 and WT tobacco plants under chronic salt treatment (200mM NaCl)

R2: After reviewing the data used for figure 3.B), we found that the percentage of chlorotic leaves for each dataset was overcounted. In order to overcome this situation, the figure was modified. An one-way ANOVA test was performed followed by a Bonferroni adjustment to determine significant differences between the mean values of each dataset and the mean value of the wild-type (WT) dataset. We also corrected the statistical differences shown for T1L4 (green and chlorotic leaves datasets) and for T1L9 (green leaves dataset).

Q3: In Figure 4B I cannot see clear difference sin survival rate at 75 mM and 100mM

R3: The mean and SD of each bar represent the results obtained from six plates, each containing seven calli (WT, EV, or DcLCYB2) subjected to different NaCl concentrations (e.g. a total of 42 WT, EV or LCYB2 calli in plates 0mM NaCl, other 42 for each gene/control in plates 75mM and so on). After 5 and 15 days, the survival rate was quantified for each individual plate. A two-way ANOVA test was performed followed by a Dunnett's multiple comparisons test to determine significant differences between DcLCYB2 and EV dataset against the WT control. We revised the data an analysis and we are still getting the same results.

Reviewer 2 Report

The authors are addressing carotenoid biosynthesis in the orange storage root of carrot. Carotenoid biosynthesis is well characterized in many species, but concerning carrot, there are some gaps in information.

The manuscript concentrates on lycopene β-cyclase (LCYB) that is necessary for biosynthesis of both α- and β-carotenoids. A leaf abundant LCYB (DcLCYB1) has been characterized in carrot but there is indication also of a root abundant form (DcLCYB2). However, the gene for this enzyme has been annotated as capsanthin-capsorubin synthase (CCS) and a section in Discussion explains why this may happen in automatic annotation.

The task of this work was to show that DcLCYB2 indeed is an LCYB, and is localized in the chloroplast. They further show that as with other LCYB genes, ectopic expression leads to increased salt tolerance and has therefore potential in GM breeding of crop plants for increased salt tolerance.

The paper is very well written and a pleasure to read. Few minor remarks:

Line 420: GV3101 is an empty agrobacterium strain, cured for its Ti-plasmid. In order to perform gene transfer, the strain has to contain a disarmed Ti-plasmid that provides the vir-functions. This should be mentioned in methods. Typically the strain often erroneously referred to as GV3101 is GV3101(pMP90), but it could be something else as well.

Line 452: The extraction procedure is written as if water and acetone would make two phases, but they are completely miscible.

Line 497: Tobacco species should be mentioned. It is usually N. benthamiana for agroinfiltration.

Title: The term carotenoid cleavage dioxygenase (CCD) is apparently wrong, should be capsanthin-capsorubin synthase (DcCCS).

Author Response

The task of this work was to show that DcLCYB2 indeed is an LCYB, and is localized in the chloroplast. They further show that as with other LCYB genes, ectopic expression leads to increased salt tolerance and has therefore potential in GM breeding of crop plants for increased salt tolerance.

C1: The paper is very well written and a pleasure to read.

R: We sincerely appreciate reviewer`s comment

Q1: Line 420: GV3101 is an empty agrobacterium strain, cured for its Ti-plasmid. In order to perform gene transfer, the strain has to contain a disarmed Ti-plasmid that provides the vir-functions. This should be mentioned in methods. Typically, the strain often erroneously referred to as GV3101 is GV3101(pMP90), but it could be something else as well.

R1: Indeed the strain is the GV3101::pMP90. It was corrected.

Q2: Line 452: The extraction procedure is written as if water and acetone would make two phases, but they are completely miscible.

R2: Thank you, it was fixed in the manuscript.

Q3: Line 497: Tobacco species should be mentioned. It is usually N. benthamiana for agroinfiltration.

R3: Yes, you are right. We specified that it was in N. benthamiana.

Q4: Title: The term carotenoid cleavage dioxygenase (CCD) is apparently wrong, should be capsanthin-capsorubin synthase (DcCCS).

R4: It was an awful mistake! Thank you very much to notice it